# Emergy as a Tool to Evaluate Ecosystem Services: A Systematic Review of the Literature

**Ana Carolina V. Nadalini** [1,*], **Ricardo de Araujo Kalid** [2] **and Ednildo Andrade Torres** [3]

1   Energy and Environment Interdisciplinary Center (CIENAM), Federal University of Bahia (UFBA), Salvador 40170-115, Brazil
2   Federal University of the South of Bahia (UFSB), Itabuna 45613-204, Brazil; ricardo.kalid@gmail.com
3   Polytechnic School, Federal University of Bahia (UFBA), Rua Aristides Novis, 2, 3rd Floor, Salvador 40170-115, Brazil; ednildotorres@gmail.com
*   Correspondence: acnadalini@hotmail.com; Tel.: +55-79-99927-7300

**Abstract:** The objective of this paper is to present a review of current research on the valuation of ecosystem services, using emergy evaluation methodology (EME). A bibliometric analysis and a systematic review were carried out between 2000 and 2020, using all of Web of Science database subfields that collected 187 papers, selected through the keywords "emergy" and "ecosystem services". In the second part of the research, we carried out a new search on Web of Science of the 187 initial articles produced, with the words "valuation" and "economic", in order to analyze those directly related to the evaluation of ecosystem services. The results showed that the EME method is an effective tool to evaluate ecosystem services, since it relates economic and ecological aspects in the evaluations. The research also indicated that the use of isolated methods does not appear to be the most appropriate solution, and that emergy used in combination with other methodologies can be used to obtain more accurate and comprehensive results to evaluate natural resources.

**Keywords:** emergy; ecosystem services; evaluation; systematic review; emergy analysis

## 1. Introduction

When *The Limits to Growth* [1] was published by the Club of Rome in 1972, questions were raised about the depletion of natural resources, as a consequence of the speed at which humanity had been consuming them. More than four decades later, we now realize that many of the issues raised are still legitimate, and that the excessive use of natural resources causes many problems. In a new study, Meadows et al. [2] concluded that humanity is dangerously in a state of overshoot. Biodiversity and ecosystem services cannot be treated as inexhaustible or infinite resources, and their value to maintain human well-being, and the costs arising from their loss and degradation, should be accounted for in some way [3–5].

With the study published by Costanza et al. [6], and the United Nations' study entitled 'The Millennium Ecosystem Assessment' [7], there was a growing interest in the evaluation and management of ecosystem services. Costanza et al. [8] argued that for a better discussion and evaluation of ecosystem services, definitions should be standardized, so that people knew what to evaluate, and were still able to find a correlation between nature, society, and the economy. Further initiatives have also emerged, such as The Economics of Ecosystems and Biodiversity (TEEB), The Common International Classification of Ecosystem Services (CICES), The Final Ecosystem Goods and Services Classification System (FEGS), and National Ecosystem Services Classification System (NESCS), the last two developed by the U.S. Environmental Protection Agency.

Ecosystem services were then defined as the direct and indirect benefits that people obtain from ecosystems, and a classification of these services was developed to demonstrate their importance to human well-being, which was not valued in the traditional

economic view. In this sense, the evaluation of natural resources has been a recurring theme in scientific research [8,9], and two types of approach are possible: (1) an evaluation focused on the economic viewpoint, where ecosystem services are evaluated in monetary units (willingness to pay, market prices, and replacement cost method, among others); (2) methods based on a biocentric viewpoint, through quantitative empirical analyses, where emergy and other methods such as exergy, the ecological footprint (EF), life cycle analysis (LCA), and material flow analysis are included.

Methods focused on the anthropocentric viewpoint treat the environment as an appendix to the economy, where the services provided by ecosystems are seen as a means to quantify biodiversity in economic terms, but do not consider the contribution of nature and its formation of the raw material used, nor the damage generated by the future exhaustion of the natural resource, nor the expenses resulting from the social exclusion of local communities [5,6,10,11].

In the biocentric viewpoint, the ecosystem is seen as a whole, and the economy is only a part of it. Therefore, the entire economic system is under the laws of physics, and, more specifically, the Second Law of Thermodynamics. The main author defending this idea was Georgescu-Roegen [12] who stated that economics is a thermodynamic process, which transforms low entropy materials into high entropy materials and, consequently, deteriorates the material base on which it is irreversibly established (creating entropy), which means that the role of natural resources (the material basis) should be acknowledged in the economy.

Emergy evaluation methodology (EME) is a biocentric method based on thermodynamics developed by H. T. Odum in 1983. Emergy (written with an "m" as a reference to energy memory) can be defined as the sum of the energies available of all types to produce a product or service, expressing all values on the same common basis [13]. All mass, energy, money, or information flows (kg, J, $, bites) are transformed into solar emergy (seJ), using a conversion factor, formerly called transformity, known as the Unit Emergy Value (UEV), expressed in solar energy joules (seJ/J), determined by the annual direct emergy flow required for the maintenance of the biosphere, considered as the basis for calculation for the emergy baseline, and has been revised to $12.00 \times 10^{24}$ seJ/year by Brown and Ulgiati [14,15].

Despite its conceptual and methodological complexity, and the debate on the use of economic and biophysical values, the absence of a more practical tool that can be used in the "real world" has justified the use of the emergy approach by universities and public agencies, as a tool to assist in decision-making on the use of natural resources [16,17].

Two recently published emergy related reviews [18,19], have mainly demonstrated the evolution of the use of emergy as a tool to investigate research trends in natural resource assessments. Chen et al. [20,21] and He et al. [22] pointed out the need of integrating different methods to study the environment, but more research on this subject should be explored. In that respect, this paper aims to investigate scientific publications that use emergy analysis with other methodologies as a tool to evaluate ecosystem services that include social, economic, and environmental dimensions. A bibliometric analysis and a systematic review were carried out between 2000 and 2020, using all of Web of Science database subfields that collected 187 papers, selected through the keywords "emergy" and "ecosystem services" and language = "English". In the second part of the study, we refined our research on the 187 initial articles, and found 66 articles that directly related to the evaluation of ecosystem services in monetary and biophysical units. Thus, we carried out a more detailed analysis of the methods used.

Following this introduction, we present the main concepts of ecosystem services in order to format the theoretical framework. Section 2 describes the main research methods. The results and discussions are presented in Sections 3 and 4, respectively. Lastly, Section 5 indicates the most important conclusions and research contributions.

*Ecosystem Services and EME*

In order to evaluate an ecosystem, its components and the relationship between them are quantitatively described. For this, the ecosystem is considered a system governed by the laws of thermodynamics; they are open systems, with a hierarchy that is organized and self-regulated. In order to quantitatively understand the functioning of an ecosystem, the incoming and outgoing of energy and matter that pass through its borders into and from the environment are accounted for. The approaches that analyze ecosystem flows are called "the donor-side approach" and, according to Pulselli et al. [23], the EME method is the most commonly used.

In the EME method, the entries are considered the ecosystem functions or natural capital which can be defined as the stock of resources provided by nature (donor side), and the outputs, the ecosystem services and, therefore, it is essential to define and differentiate the terminologies used in the evaluations. Ecosystem services are now considered a complement to the emerging flows within the ecosystem. The concept of ecosystem services is anthropocentric, and services only exist since there is an end user with great influence on the dynamics of the ecosystem, depending on use [23,24].

In the definition of ecosystem services [6,7,25], the user approach is applied; in other words, the services that come out of the ecosystem are evaluated and, therefore, are called the user side approach, from an anthropocentric viewpoint. Therefore, in these evaluations, the system users must be defined, to consider which outputs should be evaluated, since each one may describe the same system in different ways. It is also important to consider that the ecosystem not only provides services for humans, but also sustains the functions, dynamics and balance between other species throughout the ecosystem.

The Millennium Ecosystem Assessment [7] proposed a classification system for ecosystem services that was adapted by TEEB [25], and has been used in assessments to emphasize the interaction between the natural, social, manufactured and human capital required to produce the following services:

1.  provision services: ecosystem services which combined with manufactured, human and social capital produce food, firewood, and fibers etc.;
2.  regulation services: flood control, water regulation, air quality, pollination and climate control;
3.  cultural services: provide recreation, cultural identity, and landscape esthetics, among other cultural benefits;
4.  support services: are characterized by basic ecosystem processes, such as soil formation, nutrient cycling and habitat provision, and are the services required to maintain the first three services.

In order to clarify the relationship between emerging flows and ecosystem services, a broader and transdisciplinary perspective [8,26] should be adopted. Since ecosystem services are the contribution of natural capital to human well-being, the evaluation of ecosystem services must include all the values of provision, regulation, cultural, and support services.

One of the main disadvantages of biocentric evaluations, such as emergy analysis, is the fact that a common unit of measurement may not be easily associated with all types of services and may underestimate the value of one or more ecosystems. These evaluations also tend not to calculate the economic value of ecosystem services. A recognition of this disadvantage prompted us to investigate whether the emergy theory could be used to evaluate ecosystem services by overcoming these limitations.

## 2. Materials and Methods

### 2.1. Bibliometrics Analysis

Bibliometric research, which was used in this article, is a scientific research methodology utilized in various fields of science to map publication patterns on a given subject, through mathematical and statistical database analyses. Quental and Lourenço [27] argue that bibliometric analysis is one of the most significant sources of information to evaluate

the influence or credibility of articles, research institutions, or researchers, through research which is more focused on content analysis and the evolution of scientific paradigms. Our research was developed in order to broaden the knowledge of publications related to the evaluation of ecosystem services obtained by EME, through a study of articles on the subject.

According to Chen et al. [21], bibliometric research can identify the main topics related to the research theme, and the emergence of new topics. It can also point out publication trends map the productivity of authors, organizations, and countries, and guide future research, and with the use of keywords, it has proven to be effective for emergy-related research.

A bibliographic survey was conducted from October 2020 through January 2021 on the Web of Science scientific database, which is multidisciplinary and indexes the most cited journals in their respective areas, providing tools for the analysis of citations, references, the h index, and other bibliometric analyses. In addition, the Web of Science is recognized around the world as one of the main sources of information, both in the academic context and other bibliometric studies [21].

### 2.2. Selection of Terms and Timeframe

In this study, we selected all of the Web of Science database subfields, including: The Science Citation Index Expanded, Social Sciences Citation Index, Conference Proceedings Citation Index-Science, and Conference Proceedings Citation Index-Social Science & Humanities. The database search resulted in 187 papers, selected through the keywords "emergy" and "ecosystem services", language = "English", and type of papers = "articles", "proceeding papers", "reviews" and "editorial material", in the Web of Science topic criteria (title, abstract, author keywords) published between 2000 and 2020. In the second part of the research, we carried out a new search on Web of Science of the 187 initial articles produced, with the words "valuation" and "economic", in order to analyze those directly related to the evaluation of ecosystem services. A systematic literature review of scientific articles was conducted following the work of Moher et al. [28] using the PRISMA model (Preferred Reporting Items for Systematic Reviews and Meta-Analyses), along with an adaptation of the "snowballing" technique [29], where we used our knowledge and judgment to decide on the inclusion of new terms from the list of full text papers which were found initially.

The selection of the year at the beginning of the research was based on Chen et al. [21], who pointed out that "Environmental Accounting: Emergy and Environmental Decision Making", published in 1996, was a milestone that triggered the start of research in this emerging area. Although the initial research year was 1996, publications only appeared from 2000. This can be explained by the publication of a provocative study on economic value associated with the use of ecosystem services calculated on a global scale by Costanza et al. in 1997. Coscieme et al. [30] and de Groot et al. [4] argue that the number of articles and publications on the monetary valuation of natural resources, ecosystem services, and biodiversity increased on account of this publication.

### 3. Results

#### 3.1. Number of Publication and Citations

Of the 187 publications researched, 89% correspond to articles. There was not a significant number of other types of documents, which were as follows: proceeding papers (5.9%), review papers (5.3%), editorial material (2.1%), and corrections (1.6%). As shown in Figure 1, from 2005, the number of publications and citations increased more rapidly, from 1 publication in 2005, to 27 in 2020, with a peak of 34 in 2019, indicating increased interest in the evaluation of ecosystem services with the use of EME in the last two decades.

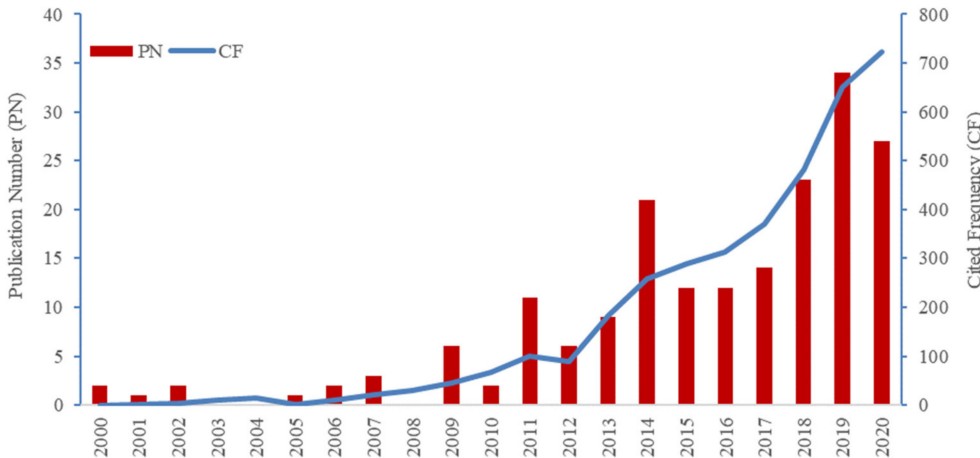

**Figure 1.** The performance of selected publications between 2000 and 2020. Publication number (PN) and cited frequency (CF) of papers published.

*3.2. The Performance of Different Journals*

Publications related to the analysis of ecosystem services using EME were published in 70 journals between 2000 and 2020. To complement the analysis of the publications, we decided to examine the impact factor (IF) of the 10 most productive journals, which included 61% of the total publications analyzed, as shown in Table 1.

**Table 1.** Performance of the top 10 most productive journals between 2000 and 2020.

| Journal Title | Numbers of Papers | Percentage of Total | Impact Factor (IF) |
|---|---|---|---|
| Journal of Cleaner Production | 28 | 14.89% | 6.395 |
| Ecological Modelling | 25 | 13.30% | 2.363 |
| Ecological Indicators | 19 | 10.11% | 4.490 |
| Ecosystem Services | 11 | 5.85% | 5.572 |
| Science of The Total Environment | 9 | 4.79% | 5.589 |
| Sustainability | 9 | 4.79% | 2.075 |
| Ecological Engineering | 7 | 3.72% | 3.406 |
| Journal of Environmental Accounting and Management | 5 | 2.66% | 0.630 |
| Agricultural Systems | 4 | 2.13% | 4.131 |
| Environment Science and Pollution Research | 3 | 1.60% | 3.306 |

For Thomaz et al. [31], IF is the most efficient bibliometric index to evaluate the quality of a journal, and its value is determined by the publications and citations of articles published by the journal in the last two years, calculated by the Institute of Scientific Information (ISI). Of the top 10 journals analyzed, all are relevant publications in the area of ecosystem services and emergy, except for the Journal of Environmental Accounting and Management (8th position). The Journal of Cleaner Production has the highest impact factor (6395), followed by Science of the Total Environmental (5589) and Ecosystem Services (5572). The four largest journals correspond to almost half the publications and show that papers including the theme of ecosystem services and EME have been published in journals of great scientific relevance in the last two decades. These results corroborate those found by Chen et al. [21] and Chen et al. [20], when they researched the emergy theoretical framework until 2014.

*3.3. Country Performance and Academic Collaboration*

Table 2 shows the ten most productive countries in relation to publications involving the theme of ecosystem service evaluations and EME. A total of 35 countries published articles on the subject between 1996 and 2020, with China publishing the most (93 articles]), followed by Italy (57 articles) and the United States (44). It can be argued that the main

countries with publications on the theme researched are developed, generally consume more natural resources, or have more scarcity due to the size of the territory (e.g., Luxembourg) and, therefore, may have more interest on the topic of the evaluation of services produced by ecosystems, which directly influence the well-being of the population. In this sense, China published almost half the articles surveyed, demonstrating its growing interest in natural resources, but according to Chen et al. [21] this great increase in publications might also be attributed to the large-scale initiatives on basic research in China, initiated in 1995.

**Table 2.** Performance of the top 10 most productive countries between 2000 and 2020.

| Country | Total Publications | Percentage of Total |
| --- | --- | --- |
| China | 93 | 49.5% |
| Italy | 57 | 30.3% |
| USA | 44 | 23.4% |
| Brazil | 22 | 11.7% |
| Luxembourg | 9 | 4.8% |
| France | 7 | 3.7% |
| Spain | 7 | 3.7% |
| Sweden | 7 | 3.7% |
| Australia | 5 | 2.7% |
| Denmark | 5 | 2.7% |

The institutions with the most publications follow the same trend as the countries researched, as shown in Table 3. The research leader was the Chinese Benjing Normal University, followed by the University of Naples, Italy (2nd place), and a second Chinese institution, the Chinese Academy of Sciences (3rd place). It should be noted that although Brazil does not have a shortage of natural resources, it is interested in their export, and ranked 4th in terms of publications. The Brazilian institution with the most publications was the Paulista University (UNIP) since it has a center for emergy studies.

**Table 3.** Performance of the top 10 most productive institution between 2000 and 2020.

| Institute | Total Publications | Country |
| --- | --- | --- |
| Beijing Normal University | 40 | China |
| Parthenope University, Naples | 40 | Italy |
| Chinese Academy of Sciences | 21 | China |
| Universidade Paulista | 14 | Brazil |
| State University System of Florida | 12 | USA |
| United States Environmental Protection Agency | 12 | USA |
| University of Florida | 12 | USA |
| Beijing Engn Res Ctr Watershed Environm Restorat | 11 | China |
| Shanghai Jiao Tong University | 11 | China |
| University of Siena | 10 | Italy |

*3.4. Author Performance and Most Cited Papers*

According to the database, approximately 500 authors have published papers on ecosystem assessment using EME, with an average of 2.6 authors per article. Following the same pattern as the publications found by Chen et al. [21] and Chen et al. [20], a small group of authors contribute to the majority of the publications, where the 10 most productive authors contributed to almost 70% of the publications on emergy and ecosystem assessment (Table 4).

**Table 4.** Performance of the top 10 most productive authors between 2000 and 2020.

| Author | Country | Total Publications | Percentage of Total |
|---|---|---|---|
| S. Ulgiati | Italy | 22 | 11.7% |
| B.F. Giannetti | Brazil | 14 | 7.4% |
| P.P. Franzese | Italy | 14 | 7.4% |
| G.Y. Liu | China | 14 | 7.4% |
| E. Buonocore | Italy | 11 | 5.8% |
| Geng Y | China | 10 | 5.3% |
| Almeida C.M.V.B. | Brazil | 9 | 4.8% |
| B. Rugani | Luxemburg | 9 | 4.8% |
| Agostinho F. | Brazil | 8 | 4.3% |
| S. Bastianoni | Italy | 8 | 4.3% |

The authors with the most publications were S. Ulgiatti with 22 articles, followed by B. F. Giannetti and B. P. Franzese, both with 14 articles. Some of the most productive authors were Odum's students, such as S. Ulgiatti, and S. Bastianoni [21]. Three of the main authors are Brazilian researchers who are professors at the Paulista University (UNIP), and have been researching EME and ecosystem services with their most recent joint publication being on the evaluation of urban parks in the city of São Paulo [32]. In this publication, it is interesting to note that G.Y. Liu is also one of the authors. The presence of several Chinese authors is observed in our research, which corroborates the interest of Chinese academia on the subject, confirmed by the analysis by country shown in Table 2.

The ten most popular articles have been cited more than 800 times since their initial publication until the end of 2020. The most cited article was "Accounting for Ecosystem Services in Life Cycle Assessment, Part I: A Critical Review", published in *Environmental Science & Technology* in 2010. The other 9 most cited articles are listed in Table 5 [33–40].

**Table 5.** Performance of the top 10 most cited articles between 2000 and 2020.

| Title | Year | Citations | Author | Journal |
|---|---|---|---|---|
| Accounting for Ecosystem Services in Life Cycle Assessment, Part I: A Critical Review | 2010 | 131 | Zhang, Y.; Singh, S.; Bakshi, B. R. | Environmental Science & Technology |
| Effects of River Impoundment on Ecosystem Services of Large Tropical Rivers: Embodied Energy and Market Value of Artisanal Fisheries | 2009 | 126 | Hoeinghaus, D. J.; Agostinho, A. A.; Gomes, L. C.; Pelicice, F. M.; Okada, Edson K.; Latini, J. D.; Kashiwaqui, E. A. L.; Winemiller, K. O. | Conservation Biology |
| A modified method of ecological footprint calculation and its application | 2005 | 114 | Zhao, S; Li, ZZ; Li, WL | Ecological Modelling |
| Obscuring Ecosystem Function with Application of the Ecosystem Services Concept | 2010 | 98 | Peterson, M. J.; Hall, D. M.; Feldpausch-P., Andrea M.; Peterson, T. R. | Conservation Biology |
| A thermodynamic framework for ecologically conscious process systems engineering | 2002 | 83 | Bakshi, BR | Computers & Chemical Engineering |
| The energetic basis for valuation of ecosystem services | 2000 | 83 | Odum, HT; Odum, EP | Ecosystems |
| The value of the seagrass Posidonia oceanica: A natural capital assessment | 2013 | 77 | Vassallo, P.; Paoli, C.; Rovere, A.; Montefalcone, M.; Morri, C.; Bianchi, C.N. | Marine Pollution Bulletin |
| Ecosystem services assessment: A review under an ecological-economic and systems perspective | 2014 | 76 | Hayha, T.; Franzese, P. P. | Ecological Modelling |
| Improvements to Emergy Evaluations by Using Life Cycle Assessment | 2012 | 69 | Rugani, B.; Benetto, E. | Environmental Science & Technology |
| A combined tool for environmental scientists and decision makers: ternary diagrams and emergy accounting | 2006 | 66 | Giannetti, BF; Barrella, FA; Almeida, CMVB | Journal of Cleaner Production |

In addition to the publications researched on the Web of Science database, according to Chen at al. [21] there are other publications that are equally important and frequently cited, such as the Proceedings of the Biennial Emergy Conference organized by Prof. Mark T. Brown of the University of Florida, and material on the National Environmental Accounting Database (NEAD). We also verified that three of the most cited articles use other research tools, in conjunction with EME, to evaluate ecosystem services, such as LCA [39,41] and the ecological footprint—EF [34]. He et al. [22] also verified a significant growth in the use of emergy analysis with other methodologies from 2007. For this reason, in the next section, we analyzed the methodologies used in the main researched articles that incorporate the emergy approach, along with other methodologies specifically to evaluate ecosystem services.

## 4. Discussion

As explained in Section 2, in the second part of the study, we refined our research on the 187 initial articles, to analyze the articles directly related to the evaluation of ecosystem services. We found 72 articles whose results were calculated in monetary and biophysical units. Of this total, we found that 6 articles were bibliographic reviews of EME, with the joint application of other methodologies, which were then excluded from the analysis. Thus, from the remaining 66 articles, we carried out a more detailed analysis of the methods used. Of this total, it was verified that the exclusive use of the emergy approach, whose ecosystem service values are obtained in dollars, accounts for the majority of the studies surveyed (approximately 60%) in the period between 2007 and 2020, as shown in Figure 2a. In the emergy methodology, the physical units, known as the Unit Emergy Value (UEV), expressed in solar energy joules (seJ/J), are transformed into monetary values by calculating the Emergy to Money Ratio (EMR) indicator, used to estimate the economic equivalent of emergy in dollars (in $) [13].

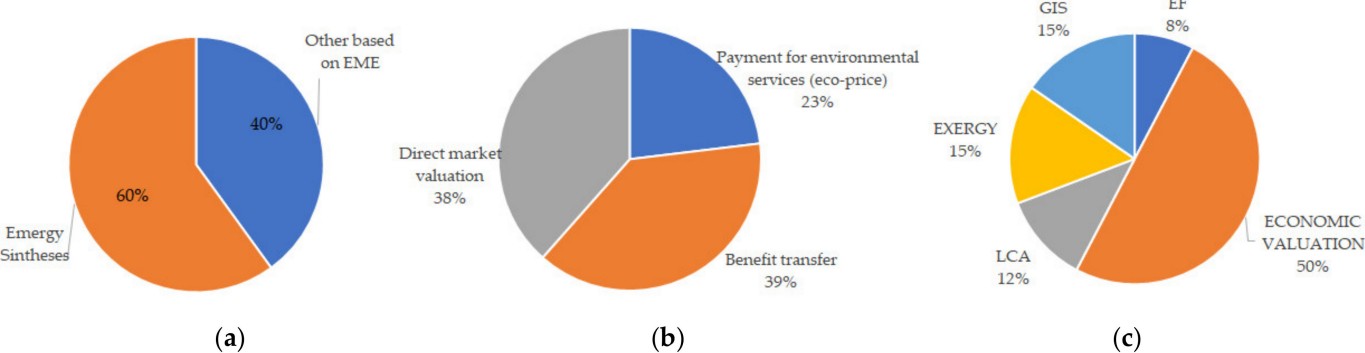

(**a**)　　　　　　　　　　(**b**)　　　　　　　　　　(**c**)

**Figure 2.** Methodologies used in the evaluation studies of ecosystem services in the period between 2000 and 2020. (**a**) Valuation approach in ecosystem services (**b**) Details of methods used which are integrated with EME (**c**) Details of the economic valuation methodology with EME.

The estimation of ecosystem services in dollars has been used by government agencies as a subsidy to formulate public policies. Campbell and Brown [42] estimated the value of natural capital and ecosystem services in nine regions of the U.S. Forest Service. Campbell and Ohrt [43] evaluated the state of Minnesota as part of the U.S. EPA's environmental policy project. The EMR is calculated through the ratio between the emergy flow of a country in relation to its GDP, expressed in seJ/$, and represents how much emergy corresponds to a unit of money produced by the national economy [44].

Almeida et al. [32], used the concept of environmental accounting in emergy to evaluate 73 parks in the city of São Paulo in Brazil, establishing a cost–benefit ratio between the production of environmental services and municipal investment, in order to assist with urban park management. Emergy indices, such as EMR, have been used to evaluate some of the ecosystem support, regulation, and provision services. Tilley and Swank [45]

evaluated the multiple functions of forest systems; Zhao and Wu [46] calculated the annual values of a mangrove ecosystem in China; Gianetti et al. [47] evaluated the ecosystem services of a native forest area within a coffee farm in Brazil; and Vassalo et al. [17] and Franzese et al. [48] estimated the value of a marine ecosystem in protected areas.

Since the values of ecosystem services can be expressed in biophysical and monetary units, the combination of emergy analysis with other methods has been widely used and represented 40% of the articles surveyed. Our research corroborates the findings of He et al. [22], which observed a significant increase in the number of articles that used the integration of EME with other methods between 2007 and 2018. The methods analyzed in this study vary in relation to the evaluation purpose (valuation, sustainability, and support capacity) but all have the systemic approach in common. Among these methods, the most commonly used were economic evaluations in 50% of the articles, with the economic approach that focuses more on the economic value of natural resources (benefit transfer, market value and payment for environmental services), followed by other methodologies more centered on biophysical aspects, such as exergy (15%), LCA (11%), GIS (15%), and EF (11%), as shown in Figure 2b We will now list the main aspects of each methodology.

### 4.1. Economic Valuation Approach and EME

The use of economic evaluation using EME has been increasingly present in the articles researched, due to the proximity to monetary units that facilitate an understanding and comparison of the results. As pointed out by Costanza [49], if at least one of the components being trade-off is expressed in monetary units, ecosystem services can be expressed in those units as well.

Among the publications that used some type of monetary unit to evaluate ecosystem services, as shown Figure 2c, the most commonly used was the benefit transfer method, with 39%, followed by market evaluation (38%) and payment for environmental services (23%).

The benefit transfer method consists of transferring the services estimated in previous studies conducted in different locations, but with similar physical characteristics [50]. With the objective of evaluating the relationship between environmental services and their monetary value, Pulsielli et al. [23] have calculated the ratio between the emergy flow that supports the entire biosphere in relation to the value of global ecosystem services calculated by Costanza et al. [6], obtained through a specific environmental conversion factor called the "environmental emergy-to-money ratio" (EEMR).

In a second study published by Coscieme et al. [30,51], the global values calculated for the biosphere were recalculated on a smaller scale for the emergy evaluation of 16 biomes, and their values represent a type of redistribution of the classic values of ecosystem evaluations once the inputs of renewable resources required for these biomes to exist are computed, so they have a higher order of magnitude. Berrios et al. [52] used the methodology proposed by Pulselli et al. [23] to estimate the total marine benthic ecosystem services on which the regional economy of three bays, located in northern Chile, was based. The total dollar value found (em$0.48 m$^{-2}$y$^{-1}$) for the coastal ecosystem was approximately 8.2% of that found in hypothetical dollars (US$5.87 m$^{-2}$y$^{-1}$). These results show the importance of calculating values closer to the economic reality of society. So that effective protection measures for the use of natural resources, can be taken.

In another methodology proposed by Campbell and Tilley [53], emergy analysis was used to establish a payment system for environmental services, based on a monetary unit called the "eco-price". The eco-price is based on the collection of amounts that were paid by society to avoid loss, or to restore damage caused to a given environmental resource, in the form of a monetary unit ($) paid for environmental services provided by ecosystem services. The eco-price is calculated using the ratio between emergy and monetary values, such as the marketed value of a ton of wood, which must be calculated specifically for each region studied. Using this methodology, Campbell and Tilley [54] and Campbell [55]

evaluated the annual benefits from ecosystem services in Maryland forests in the United States at $5,767 per hectare of forest and $9,693 per hectare of fresh water.

### 4.2. Integration of EME with the Ecological Footprint, Life Cycle Assessment, Exergy and GIS

According to Yu et al. [18] and Zhong et al. [19], there is a tendency to use analyses that combine EME with other approaches, such as LCA and EF, to determine the efficient use of a resource or the environmental efficiency of a process. Exergy and EF use coefficients to transform the inputs of a system into the same unit that can be used as LCA entries. For this reason, these approaches are usually integrated, such as Arbault et al. [56] who used EF, LCA, and emergy; Sheng et al. [57] used the concepts of exergy and EF in emergy analysis, and Meng et al. [58] used exergy and LCA, based on EME methodology. A more detailed analysis of the potential use of LCA and emergy can be found in Rugani and Benetto [39] and Wang et al. [59]. McDougall et al. [60] and Yang et al. [61] used indicators obtained through EME and the ecological footprint in agriculture systems with an urban impact. Similar to emergy, one of the advantages of exergy is to present several values in one measure, which is generally used to evaluate non-renewable inputs, such as metal and fossil fuels, among others.

Another concept is eco-exergy, which reflects energy stocks which are currently available, while emergy quantifies the historical memory of development and maintenance in the system network. Therefore, the proportions of eco-exergy to emergy (Ex/Em) and eco-exergy to empower (Ex/Em) are informative indicators for the efficiency with which a system uses external energies to form and maintain its own stability, thermodynamic structure and information content. Lu et al. [62] calculated these indicators to evaluate the ecosystem health and sustainability of three forests in China.

EF represents the impact of human activities in terms of land area. Together with emergy analysis, biocapacity is calculated and compared in terms of global emergy density, based on renewable resources available (sun, wind, chemical energy in the rain, geopotential energy in the rain, and earth cycle energy). Yang et al. [61] evaluated the eco-elliptical deficit of ecosystem services in some provinces of China, with a 100 year projection, using EME and EF. In the evaluation of ecosystem services, the EF evaluation is incomplete, since the methodology can only account for biological goods and services, such as crops, fish and wood. Many essential services are not included, since not all of them can be converted to a common unit, such as the value per area [41].

Some authors have started to integrate EME and the geographic information system (GIS), as a way of highlighting the importance of the spatial dimension of natural resources, and to assist in decision-making for the preservation of natural resources. Mellino et al. [63] evaluated natural and human capital by mapping southern Italy, demonstrating the importance of preserving areas with a lower economic value, but with high degradation or stocks of natural resources. Pu et al. [64] used GIS techniques to obtain an emergy index in order to evaluate a mountainous area in China with difficult access, demonstrating the importance of remote monitoring. Table 6 demonstrates the strengths and weakness of the methodologies found in the research used with EME to perform the economic evaluation of ecosystem services. Economic evaluation (also known as mathematical modelling such as market prices, benefit transfer, the eco-price and simulations), EF, LCA, exergy, eco-exergy, and GIS are prevailing methods on natural resource accounting, and when combined with emergy analysis can express ecosystem services values in biophysical and monetary units.

**Table 6.** Comparison of main ecosystem service economic evaluation methods using EME.

| Methodology | Reference | Strengths | Weakness |
|---|---|---|---|
| Economic evaluation (we consider methods that mainly contain mathematical methods such as market prices, benefit transfer, the eco-price and simulations etc.) | Lu et al. [10]; Campbell and Tilley [53]; Pulselli et al. [23] | The values obtained are easy to understand by both the general public and the public authorities. The various methodologies can be used for any type of natural resource, in any location. | The monetary value may not represent the true value of the natural resources. The inevitable estimates and transfers of values between locations increase the uncertainties of the results. |
| EF | Mancini et al. [9] Zhang et al. [41] | Excellent communication tool with the general public. | It cannot account for all ecosystem services, only those generally measured by hectares. |
| LCA | Wang et al. [59]; Rugani et al. [39]; Zhang et al. [41] | The use of emergy indicators in LCA models can improve UEV quality. The analysis process is quantitative, detailed and accurate. | Used only to assess system sustainability. The emergy focuses on the donor´s view, and the LCA on the user's view. The combination of the two methodologies has not yet received unanimous approval. |
| Exergy/eco-exergy | Lu et al. [62]; Zhang et al. [41]; Bastianoni et al. [65] | Based on thermodynamics and has been used to understand ecosystem dynamics. Both have the equivalent of solar energy as a conversion factor. | Still cannot capture differences in the quality of very different resources, such as renewable and non-renewable. The exergy calculations are much more complex. |
| GIS | Mellino et al. [63] | Useful tool for environmental planning and natural resource management. | To obtain economic values, it must be integrated with other methodologies. |

In principle, the main goal of Table 6 is not to provide a complete and detailed comparison between all the approaches, but to point out that, although these methods may differ in purposes, scopes, and data requirements, they all share the system approach in nature. Costanza [49] stated that if one element is being traded-off in monetary terms, ecosystem services can also be expressed in those units. In that sense, we could argue that economic valuation together with EME seems to be a better path to obtain a monetary value for ecosystem services, while EF and LCA are more suitable for sustainability analyses. There is an increasing consensus that no perfect method exists, nor can any single method provide a fully reliable ecosystem valuation, leading to the combination of needs with other methods providing a better understanding of any given environmental matter [66].

**5. Conclusions**

The aim of this article was to present a review of what is currently being researched on ecosystem service evaluations using EME. The results presented may help researchers in this area of study to obtain an evaluation tool that allows the joint analysis of natural and socioeconomic systems. It can also help policy makers to identify the space, time, and natural activities needed for resource production and consumption, with results that can be understood by society and, thus, generate greater added value for natural resources, and ensure the sustainability desired of their use.

In view of the growing economic development and rapid consumption of natural resources, society must understand the importance of the conservation of natural resources, and this is easier with the use of monetary values. In this sense, one should choose the most appropriate method to evaluate ecosystem services. EME is a tool to evaluate ecosystem

services, since this methodology relates economic and ecological aspects in the evaluations. As indicated in our research, the use of isolated methods has not proven to be the most appropriate solution and using EME in conjunction with other methodologies can be used to obtain more accurate and comprehensive results to evaluate natural resources.

The use of the emergy analysis represents a significant step in the ability to evaluate the services provided by the ecosystem (donor) to the human species (users) and overcome the inadequacy of single-criterion approaches. Given the transdisciplinarity of environmental issues, the use of one or more evaluation methods is required, to assess the real value of natural resources. Based on the research of 187 English-language publications obtained from the Web of Science database, we were able to obtain important aspects on the progress of research involving ecosystem service evaluations using EME between 2000 and 2020, which indicates an evolution in interest, given its recent history within academic literature.

Although an emergy assessment is based on the contribution of natural resources, calculated on the same basis, which facilitates a comparison between the various natural systems evaluated, there is uncertainty regarding the calculation and origin of the transforms, or Unit Emergy Value (UEV). A suggestion for future researchers would be to conduct case studies using EME with other methodologies to develop a scientific support database that includes this quantitative information and thus reduces the uncertainties associated with emergy evaluation.

**Author Contributions:** All authors contributed to the study conception and design. Material preparation, literature search and data analysis were performed by A.C.V.N. The other authors R.d.A.K. and E.A.T. commented on previous versions of the manuscript and critically revised the work. All authors have read and agreed to the published version of the manuscript.

**Funding:** This study was financed in part by the Coordenação de Aperfeiçoamento de Pessoal de Nível Superior—Brazil (CAPES)—Finance Code 001 as part of a PhD research.

**Data Availability Statement:** The datasets generated during and/or analyzed during the current study are available from the corresponding author on reasonable request.

**Conflicts of Interest:** The authors declare that they have no conflict of interest.

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
