# Peer review of "Emergy as a Tool to Evaluate Ecosystem Services: A Systematic Review of the Literature"

_sustainability, doi:10.3390/su13137102_

Round 1
Reviewer 1 Report
Major Revisions
The manuscript covers an interesting and quite innovative topic; it is well organised with a complete bibliography. Nevertheless, before publishing the paper, I suggest a few changes/integrations:
Introduction
Please add a very clear explanation of the objectives of your research and anticipate the method that you used to reach them;
Line 168
Please specify better the search method used. In particular, the keywords have been searched in the title? Or in the title and abstract? What kind of publications do you have considered? Only articles or also book chapters or books? Do you have included only open access articles?
Figure 1
the caption must be clear, please specify the research topic.
Lines 217 - 226
Please consider arguing better the reasoning on the distribution of paper on EME worldwide. I am not sure that the high number of publications in China depends on its growing interest in natural resources; try to add more reflections.
Lines 260 – 264
Add information on how other publications have been included in the article; what kind of method do you use?
Line 287
You introduce the concept of natural capital and environmental services, please consider providing a definition, maybe in the introduction
Line 302
Do you mean “biophysical”?
Lines 312-313
Please, before adding the abbreviation, enter the words in full
The discussion section is missing, please consider adding separately or in combination with conclusions
Conclusions
Please consider arguing more; now seems a summary of the article, try to be more critical and to propose possible further research development.
References in the text
Please make them homogeneous in style, some references are like “De Groot et al.” while others “Thomaz et al.”. Please check the editorial rules for formatting the references
References at the end of the paper
Please check the editorial rules for formatting the references according to the journal guidelines
Author Response
Point 1: Introduction :Please add a very clear explanation of the objectives of your research and anticipate the method that you used to reach them;
Response 1: I included as suggested (lines 87-95).
Point 2: Line 168 -Please specify better the search method used. In particular, the keywords have been searched in the title? Or in the title and abstract? What kind of publications do you have considered? Only articles or also book chapters or books? Do you have included only open access articles?
Response 2: I did as suggested. Please check section 2.1 Selection of terms and timeframe.
Point 3: Figure 1 - the caption must be clear, please specify the research topic.
Response 3: Done as suggested in figure 1 (lines 204-205).
Point 4: Lines 217 - 226
Please consider arguing better the reasoning on the distribution of paper on EME worldwide. I am not sure that the high number of publications in China depends on its growing interest in natural resources; try to add more reflections.
Response 4: Done as suggested (249-251).
Point 5: Lines 260 – 264
Add information on how other publications have been included in the article; what kind of method do you use?
Response 5: Done as suggested.
Point 6: Line 287
You introduce the concept of natural capital and environmental services, please consider providing a definition, maybe in the introduction
Response 6 : The concept of natural capital was included in the introduction (lines 109-110) and we switched the world environmental services to ecosystem services.
Point 7: Line 302
Do you mean “biophysical”?
Response 7: Yes, and I corrected the text.
Point 8: Lines 312-313
Please, before adding the abbreviation, enter the words in full
Response 8: Had already introduced the terms in full in the Introduction section, lines 54-55.
Point 9: The discussion section - is missing, please consider adding separately or in combination with conclusions
Response 9: Turned Section 3 into Results and Discussion. We discussed the results as being pointed out along this chapter. We also added more insights at the end of this section.
Point 10: Conclusions - Please consider arguing more; now seems a summary of the article, try to be more critical and to propose possible further research development.
Response 10: Done as suggested.
Point 11: References in the text
Please make them homogeneous in style, some references are like “De Groot et al.” while others “Thomaz et al.”. Please check the editorial rules for formatting the references
Response 11: Corrected as suggested. We also used the software Zotero with MDPI Guidelines to redo the References.
Point 12: References at the end of the paper
Please check the editorial rules for formatting the references according to the journal guidelines
Response 12: We used Zotero with MDPI Guidelines to redo the References.
Reviewer 2 Report
The research results are presented clearly and well illustrated. Although it is not innovative research, it contains many advantages that have a chance to indirectly strengthen the research in a specific scope. The authors proved that EMERGY is an effectively used tool to evaluate ecosystem services. The study is an important summary and can significantly contribute to strengthening the evaluation aspects of the issue under study.
Author Response
Reviewer 2 found the work suitable and ready to be submitted so there´s no need to respond. Thank you for taking the time to read and review it.
Round 2
Reviewer 1 Report
The revisions were made following my previous suggestions. The paper was integrated into different parts and appropriate citations, improving the quality of the writing and its understanding. The section dedicated to "discussions" is still absent and combined with that of "Results". I think it is more appropriate to keep them separate to make the text clearer and also for a stylistic issue of the Journal.
Author Response
Dear Reviewer 1,
We have made the following adjustments: divided Section 3 in two other parts as suggested in order to meet the Journal stylistic issue. In that matter, the paper shows the "Results" in section 3 and then is followed by "Discussion" in Section 4.
We hope that it meets your expecttion and thank you again for the opportunity.
Best regards,
Authors